# The Role of Echocardiography in the Management of Heart Transplant Recipients

**DOI:** 10.3390/diagnostics11122338

**Published:** 2021-12-11

**Authors:** Daniele Masarone, Michelle Kittleson, Rita Gravino, Fabio Valente, Andrea Petraio, Giuseppe Pacileo

**Affiliations:** 1Heart Failure Unit, Department of Cardiology, AORN dei Colli-Monaldi Hospital, 80131 Naples, Italy; rita.gravino@ospedalideicolli.it (R.G.); fabio.valente@ospedalideicolli.it (F.V.); giuseppe.pacileo@ospedalideicolli.it (G.P.); 2Department of Cardiology, Smidt Heart Institute, Cedars-Sinai, Los Angeles, CA 90048, USA; michelle.kittleson@cshs.org; 3Heart Transplant Unit, Department of Cardiac Surgery and Transplantology, AORN dei Colli-Monaldi Hospital, 80131 Naples, Italy; andrea.petraio@ospedalideicolli.it

**Keywords:** echocardiography, heart transplant, cardiac allograft vasculopathy, heart transplant rejection

## Abstract

Transthoracic echocardiography is the primary non-invasive modality for the investigation of heart transplant recipients. It is a versatile tool that provides comprehensive information on cardiac structure and function. Echocardiography is also helpful in diagnosing primary graft dysfunction and evaluating the effectiveness of therapeutic approaches for this condition. In acute rejection, echocardiography is useful with suspected cellular or antibody-mediated rejection, with findings confirmed and quantified by endomyocardial biopsy. For identifying chronic rejection, ultrasound has a more significant role and, in some specific patients (e.g., patients with renal failure), it may offer a role comparable to coronary angiography to identify cardiac allograft vasculopathy. This review highlights the usefulness of echocardiography in evaluating normal graft function and its role in the management of heart transplant recipients.

## 1. Introduction

Heart transplantation remains the gold-standard therapy for patients with advanced heart failure despite optimal medical treatment [1]. Over the past four decades, remarkable advances in diagnostic methods for early identification of acute and chronic rejection and in immunosuppressive therapy have resulted in a marked increase in long-term survival, with a current one-year survival rate after heart transplantation of 90% and a conditional half-life of over 13 years [2]. Echocardiography is a fundamental tool for adequately managing heart transplant recipients (HTRs), from monitoring the immediate post-operative period to surveillance of early and late post-transplant complications [3]. This review summarizes the basic principles for using echocardiography in cardiac transplant patients to allow even cardiologists not experienced in transplantation to apply this technology in the management of HTR correctly.

## 2. Normal Cardiac Allograft Structure and Function

Echocardiography is the first-line imaging modality for evaluating HTRs, providing accurate graft anatomy and function information, and should be part of all post-transplant follow-up visits. Current International Society for Heart and Lung Transplantation (ISHLT) guidelines for managing HTRs do not specify the timing of echocardiographic evaluations or recommend echocardiography as an alternative to endomyocardial biopsy in monitoring rejection [4]. However, echocardiography is routinely used when high clinical suspicion exists of acute graft rejection and for monitoring left ventricular function during episodes of acute graft rejection.

The echocardiographic evaluation of the graft is based on the assessment of the same morphological and functional parameters that are considered for the native heart. However, the results of this evaluation must be interpreted considering the pathophysiology of the transplanted heart; therefore, the main morphological and functional aspects to be considered normal in HTRs will be briefly described.

### 2.1. Left Ventricular Morphology and Function

In the first few months after heart transplantation, left ventricular mass and thickness increase due to graft edema and inflammatory cell infiltration (Figure 1) [5].

Left ventricular mass and wall thicknesses generally normalize within three months, so the persistence of left ventricular hypertrophy after this period is typically due to arterial hypertension secondary to immunosuppressive treatment and/or repeated episodes of acute rejection. Typically, the left ventricular ejection fraction and regional wall motion (except for frequent and common septal dyskinesia) are preserved in most HTRs for 10 to 15 years, unless cardiac allograft vasculopathy (CAV) develops. In addition, a decrease in left ventricular systolic function during the first year is a predictor of allograft rejection or CAV. In contrast, a late (>5 years) reduction in ejection fraction after heart transplantation correlates with CAV progression [6].

In contrast to assessment of ejection fraction, the assessment of diastolic function in HTRs is complicated by several factors and no single parameter can identify diastolic dysfunction of the graft [7]. The elevated heart rate of the denervated heart results in E-wave and A-wave fusion [8]. Pulmonary vein flows are frequently from the contraction of the recipient’s atrial tissue. Tissue Doppler imaging parameters are also altered in the graft; the e′- and a′-wave velocities are lower than those of the general population [9]. In addition, the diastolic function may be altered due to ischemia of the graft [10] or precapillary pulmonary hypertension of the recipient. Therefore, restrictive pathophysiology is common in the immediate post-transplant period and has no prognostic value. In contrast, the persistence of a significant alteration in diastolic function later after transplant [11] has a negative prognostic impact and is often related to inflammation, fibrosis, and allograft vasculopathy.

### 2.2. Right Ventricular Morphology and Function

In the early post-transplant stages, right ventricular volumes are increased due to the afterload mismatch with persistent high pulmonary pressures in the recipient. Within the first three months, a progressive reduction in right ventricular chamber size occurs, concomitant with a progressive decrease in pulmonary vascular resistance [12]. The right ventricular systolic functions of HTRs, measured by both classical echocardiographic methods (tricuspid annular plane systolic excursion (TAPSE)) and more sophisticated methods (S-wave at tissue Doppler imaging, fractional area change (FAC), and the global longitudinal strain of the right ventricular free wall) are all lower than those of healthy control participants [13]. These parameters do not tend to normalize during follow-up. However, after cardiac surgery, longitudinal parameters are known to be abnormal and not sensitive markers to assess global RV systolic function [14,15]. Despite abnormal right-sided function, in the absence of severe tricuspid regurgitation, clinical signs and symptoms of heart failure are usually not present in HTRs.

### 2.3. Atrial Morphology and Function

Atrial geometry and function are related to the surgical technique used for heart transplantation [16]. In patients who have undergone heart transplantation using the historical biatrial anastomosis, a unique morphological shape is visualized by echocardiography. This is best seen in the apical four-chamber view as an enlargement of the long-axis dimension of the atria with a ridge at the site of anastomosis (biatrial anastomosis involving the persistence of part of the recipient’s atria; Figure 2).

In contrast, in the now more standard bicaval approach, the geometry and function of the atria are comparable to those of healthy control participants. Few studies have evaluated atrial function using strain in patients with bicaval anastomosis [17]. In both atrial types of heart transplant technique, the left atrial contractile reserve is markedly reduced compared with healthy control participants [18].

### 2.4. Valve Morphology and Function

In general, the morphology of the valvular apparatus is normal in HTRs, and mild mitral and tricuspid regurgitation are common [19]. In the early post-transplantation phase, virtually all HTRs exhibit mitral valve regurgitation due to papillary muscle edema, but this is usually mild and tends to decrease over time [20]. After one year following transplantation, the most common valvular regurgitation is tricuspid, resulting from several mechanisms [21]. In the early post-transplant phases, tricuspid regurgitation is secondary to the high pulmonary pressures of the recipient; its magnitude decreases as the pulmonary vascular resistance decreases [22]. In the later stages, the leading causes of tricuspid regurgitation are the persistence of elevated pulmonary pressures, enlargement of the tricuspid annulus caused by right ventricular dilatation, and damage to the valvular and subvalvular apparatus due to frequent endomyocardial biopsies [23]. Finally, tricuspid regurgitation seems to be more prevalent with biatrial anastomosis because of altered right atrial morphology and annulus dilatation.

### 2.5. Pericardium

In HTRs, pericardial effusions are very common, particularly in the first year of follow-up [24]. After transplantation, small to moderate pericardial effusions are observed in around two-thirds of patients at three months and 25% of patients at six months [25] (Figure 3).

Severe effusions causing cardiac tamponade are rarer and usually associated with undersized hearts for the recipient’s body surface area [26]. The natural history of these effusions is variable and, in such patients, echocardiographic evaluation is required every 1–3 months to assess any changes in their extent, location, and hemodynamic impact. In cases of new-onset pericardial effusions, the main causes of effusion in HTRs should be excluded, such as right ventricle perforation due to endomyocardial biopsies, graft rejection, infection, or neoplasia.

## 3. Role of Echocardiography in the Evaluation of Primary Graft Dysfunction

Primary graft dysfunction (PGD) is the main cause of death in the first 30 days for HTRs [27]. PGD generally occurs within 24 h of transplantation, manifesting as cardiogenic shock, such as systolic blood pressure <90 mmHg for more than one hour and/or a cardiac index <2 L/min/m^2^ despite adequate right ventricular filling pressures. A diagnosis of PGD requires that these hemodynamic conditions persist despite circulatory support based on at least two inotropes/vasopressors and/or short-term mechanical assistance devices [28,29]. Furthermore, other causes of early post-operative hemodynamic compromise must be excluded to make the diagnosis of PGD, including sepsis, cardiac tamponade, bleeding, immunological processes (hyperacute rejection), or persistent severe pulmonary hypertension not responsive to pharmacological treatment [30]. In this clinical context, echocardiography plays a key role by documenting the presence of systolic dysfunction (with a left ventricular ejection fraction <45%), loss of contractile reserve, and increased right ventricular volume with systolic dysfunction (TAPSE 15 mm or a right ventricular ejection fraction <45%). In addition, echocardiography is necessary to exclude other causes of early hemodynamic compromise such as cardiac tamponade, confirm the adequate filling status of the patient, and assess for improvement in hemodynamic conditions following the initiation of supportive therapy [31].

## 4. Role of Echocardiography in the Evaluation of Acute Graft Rejection

Acute graft rejection is the leading cause of mortality during the first year after heart transplant [32]. Although post-transplant mortality due to allograft rejection has decreased significantly with the introduction of appropriate anti-rejection protocols, acute allograft rejection still occurs in approximately 15% of HTRs during the first year [33].

Immunologically, acute rejection is due to the recognition of non-self-histocompatibility antigens, resulting in the elicitation of an immune response against cardiac muscle and/or graft endothelium [34]. This immune response may be either cellular (the most common) with lymphocyte infiltration of the myocyte, with or without necrosis [35], or antibody-mediated with deposition of immunoglobulins and complement in the microvasculature [36].

Acute rejection presents with extremely variable clinical patterns, ranging from asymptomatic systolic dysfunction (due to the appearance of mild and district changes in myocardial contractility) to cardiac shock (due to total and potentially irreversible graft dysfunction). Current ISHLT guidelines indicate endomyocardial biopsy as the current standard for diagnosing and grading rejection (Table 1) [37].

However, in very few patients (0.5–1%), endomyocardial biopsy may cause complications (myocardial perforation, pericardial tamponade, tricuspid valve injury, thrombosis, or jugular vein infection) [38]. In addition, endomyocardial biopsy may be associated with sampling error and histological “false negatives”, which occur in approximately 10–20% of HTRs [39]. Therefore, in many cardiac transplant management referral centers, echocardiography is considered a first-line complementary tool in addition to biopsy for monitoring acute graft rejection.

While reduced left ventricular systolic function early after transplantation warrants endomyocardial biopsy to exclude acute rejection, the degree of left ventricular systolic dysfunction may not correlate with the severity of rejection on biopsy [40]. Other echocardiographic indicators of acute rejection, including increased left ventricular mass and wall thickness [41], changed myocardial echogenicity, and the appearance of pericardial effusion [42], occur late in the rejection process and have low sensitivity and specificity. More recent studies have sought to identify early changes in myocardial function that can reliably identify the early phase of rejection.

Abnormalities in diastolic function are the first changes to occur in acute rejection. During episodes of acute rejection, myocardial edema, immune-mediated expansion of the extracellular compartment, and interstitial fibrosis stiffen the ventricles and impair relaxation [43]. Doppler changes include a decreased pressure interval, decreased isovolumetric relaxation time, and increased E velocity [44]. However, these parameters, although very specific, have poor sensitivity due to their dependence on preload, afterload, and heart rate.

Tissue Doppler assessment is less dependent on loading conditions and cardiac frequency. Studies with limited sample sizes have shown that reduced early and late diastolic peak mitral annulus (e′ and a′) and lower systolic velocities (s′) correlate with acute rejection episodes [45]. However, although tissue Doppler measurements are sensitive to rejection, their specificity is low, reflecting the abnormal diastolic function usually present in HTRs due to the physiology of the transplanted heart [46].

Considering that graft rejection is generally a focal process, myocardial deformation imaging could, in theory, more reliably identify the small changes in segmental myocardial function typical of acute rejection. Previous studies have found that the peak systolic global longitudinal strain and strain rate are reduced in patients with acute rejection confirmed at biopsy (Figure 4).

In a study enrolling 31 HTRs, Marciniak et al. found that during episodes of acute rejection, regional strain and strain rate were reduced in the left ventricular lateral wall but not in the septum [47]. This is the result of paradoxical septal motion that happens after cardiac surgery, including cardiac transplantation. Specifically, the lateral wall peak systolic longitudinal strain was −21 ± 6% in patients with rejection grade ≤1B, compared with −13 ± 5% in patients with rejection grade >1B (*p* < 0.05). In another study, a mean systolic strain cutoff value of −27.4% predicted grade 1B or higher acute rejection with a sensitivity of 82.2% and a specificity of 82.3% [48]. Tissue Doppler imaging findings should be interpreted cautiously because strain and strain rate parameters are dependent on the transducer angle, and images can be technically challenging to acquire, with poor reproducibility. Two-dimensional speckle tracking (2D STE) overcomes the limitation of angle dependence, and this technique has also been tested in identifying acute rejection [49].

Sato et al. used 2D STE to measure left ventricular torsion to detect acute rejection. This study showed that a reduction in torsion of more than 25% from baseline could predict the presence of acute rejection with a grade 2R or higher with high specificity (95.1%) and a high negative predictive value (92.9%) [50].

A prospective study of 34 HTRs identified an independent relationship between acute rejection of grade 2R or higher and two combined echocardiographic parameters, longitudinal right ventricular free wall deformation <17% and global left ventricular longitudinal deformation <15.5%, with a negative predictive value of 98.8% [51].

Although initial experience seems to indicate that echocardiography could be a reliable tool to diagnose acute rejection, notably, the studies cited were based on different echocardiographic methodologies, enrolled small samples, and were based on the experience of single centers. Therefore, before clear and evidence-based recommendations can be made, future studies are needed, aiming at standardizing the echocardiographic strategy for surveillance of acute rejection and the role of new echocardiographic techniques based on 2D STE, such as global longitudinal strain (GLS) and myocardial work.

## 5. Role of Echocardiography in the Evaluation of Cardiac Allograft Vasculopathy

Chronic graft rejection is manifested by CAV [52]. CAV is a diffuse process that involves not only the epicardial vessels but also the coronary microcirculation [53]. The etiology of CAV is multifactorial and based on the interaction of immunological and non-immunological factors (older donor age, cytomegalovirus infection, hyperlipidemia, and type II diabetes) [54]. Unlike what normally occurs in chronic coronary syndrome, CAV, due to cardiac denervation, may progress silently before manifesting clinically as myocardial ischemia, left ventricular dysfunction or heart failure, or ventricular arrhythmia or sudden cardiac death [55,56].

Resting echocardiography provides limited diagnostic accuracy to detect CAV [57,58]. Systolic function is generally preserved even in advanced forms of CAV, making left ventricular ejection fraction an unsuitable echocardiographic marker for early detection of coronary artery disease in the transplanted heart. Wall motion abnormalities should raise suspicion of the presence or progression of CAV [59]. However, these findings are not specific because wall motion abnormalities may also develop in the absence of CAV or be due to acute rejection.

Worsening diastolic function during follow-up may also indicate CAV (as well as acute rejection), necessitating an invasive diagnostic approach [60]. For example, patients with CAV have increased durations and reduced amplitudes of both systolic and diastolic waves on tissue Doppler imaging. For example, a Doppler-derived systolic radial velocity value of ≤10 m/s showed a sensitivity of approximately 90% for angiographically and/or intravenous ultrasound-detectable CAV, but this decreased to 51% when examining stenosed major epicardial vessels, even with a cutoff of 9 cm/s [61]. Resting tissue Doppler velocities also more frequently indicate advanced stages of CAV [62].

Reduced absolute GLS values are also associated with the presence of CAV and coronary microvascular dysfunction [63]. In a population of 128 HTRs, Clemmensen et al. found a significant correlation between the extent of GLS reduction and the presence and extent of CAV [64]. Another study noted an association of reduced left ventricular circumferential deformation with the presence of coronary artery stenosis (positive and negative predictive value ≥90%, considering proximal stenosis as ≥50%) [65]. In addition, a recent study in a small cohort of heart transplant patients suggested that the absolute value of GLS and the gradient between endocardial and epicardial longitudinal strain values could be important non-invasive predictors of CAV [66].

In patients who cannot undergo repeated coronary examinations (e.g., patients with advanced renal failure), the latest ISHLT guidelines recommend the use of dobutamine stress echocardiography or treadmill stress echocardiography for CAV detection [4]. However, in HTRs, exercise may not be an adequate cardiovascular stressor due to denervation of the allograft and altered chronotropic response [67]. Therefore, in this patient population, stress echocardiography with pharmacological agents seems more favorable when this technique is indicated. Although some studies have demonstrated good sensitivity and specificity of exercise echocardiography with dipyridamole in detecting CAV [68,69], dobutamine is still the first choice. However, notably, although the use of dobutamine detects significant CAV with a sensitivity of 70–80%, it identifies mild CAV with low sensitivity [70]. The accuracy of exercise echocardiography may improve when combined with other techniques, such as myocardial deformation analysis and the use of contrast agents. For example, strain analysis may increase the sensitivity of dobutamine stress echocardiography from 63 to 88% in the detection of CAV [71], representing a promising tool warranting additional investigation.

## 6. Conclusions

Echocardiography is the first-line imaging modality of choice for the initial assessment of HTRs (Figure 5).

It is a versatile tool that provides comprehensive evaluation of the structure and function of the allograft. Echocardiographic examinations can be easily performed at the bedside in the hours after transplantation and can be serially repeated without risk or discomfort for the patient. For the identification of PGD, echocardiography is an indispensable method of documenting biventricular systolic dysfunction, required for the diagnosis of PGD, and for monitoring the effectiveness of therapy. In detecting acute rejection, traditional echocardiography, in conjunction with new echocardiographic technologies, may offer complementary insight, though it is not currently a substitute for endomyocardial biopsy. In detecting CAV, stress echocardiography has the same diagnostic accuracy as coronarography in some populations, and a normal surveillance stress echocardiogram may allow deferment of invasive coronary angiogram. More data regarding the use of innovative sophisticated tools (3D echography and myocardial work) could enhance the future role of echocardiography in the management of HTRs.

## Figures and Tables

**Figure 1 diagnostics-11-02338-f001:**
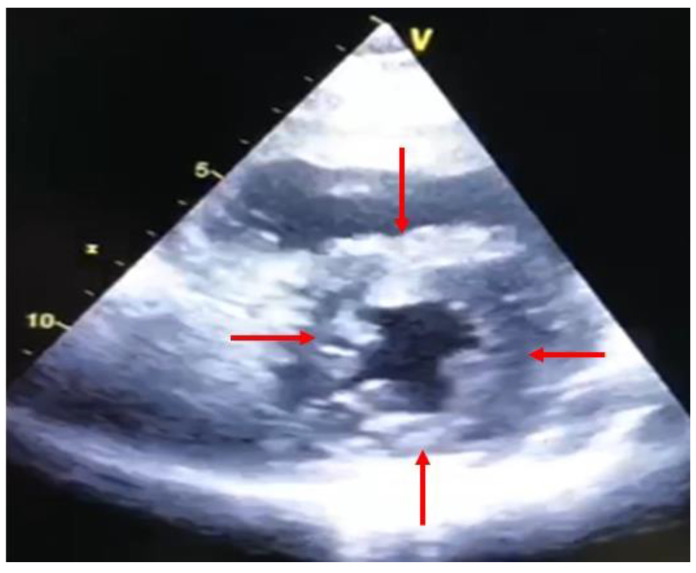
Short-axis view at trans thoracic echocardiography two months after heart transplantation. Note the concentric and symmetric hypertrophy of the left ventricle (red arrows).

**Figure 2 diagnostics-11-02338-f002:**
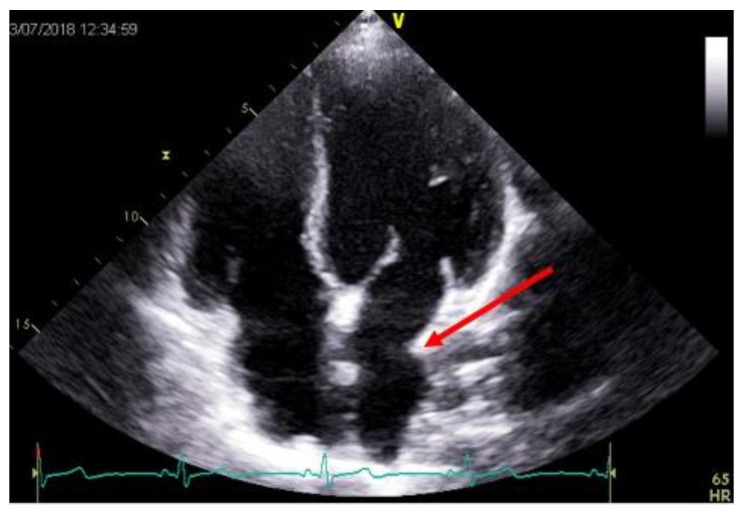
Apical four-chamber view, at trans thoracic echocardiography, after heart transplantation using a biatrial technique. Note the biatrial enlargement and the suture line in the left atrium that denotes the anastomosis between the donor and recipient atria (red arrow).

**Figure 3 diagnostics-11-02338-f003:**
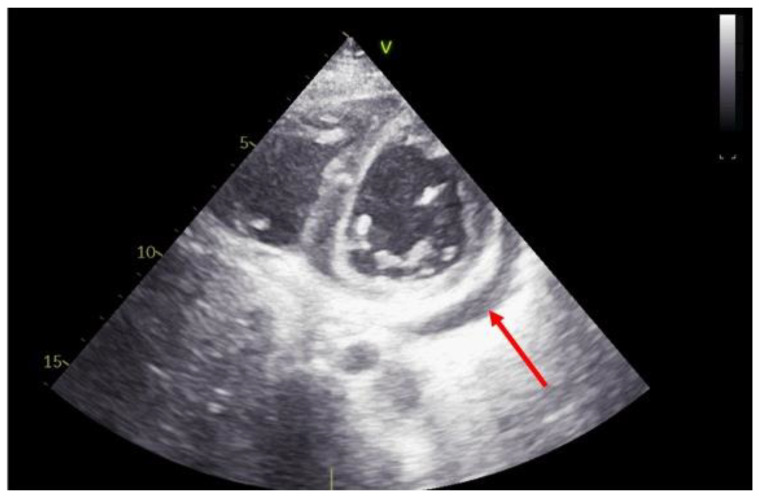
Short-axis view at trans thoracic echocardiography, one month after heart transplantation. Note the small pericardial effusion in the lateral position (red arrow).

**Figure 4 diagnostics-11-02338-f004:**
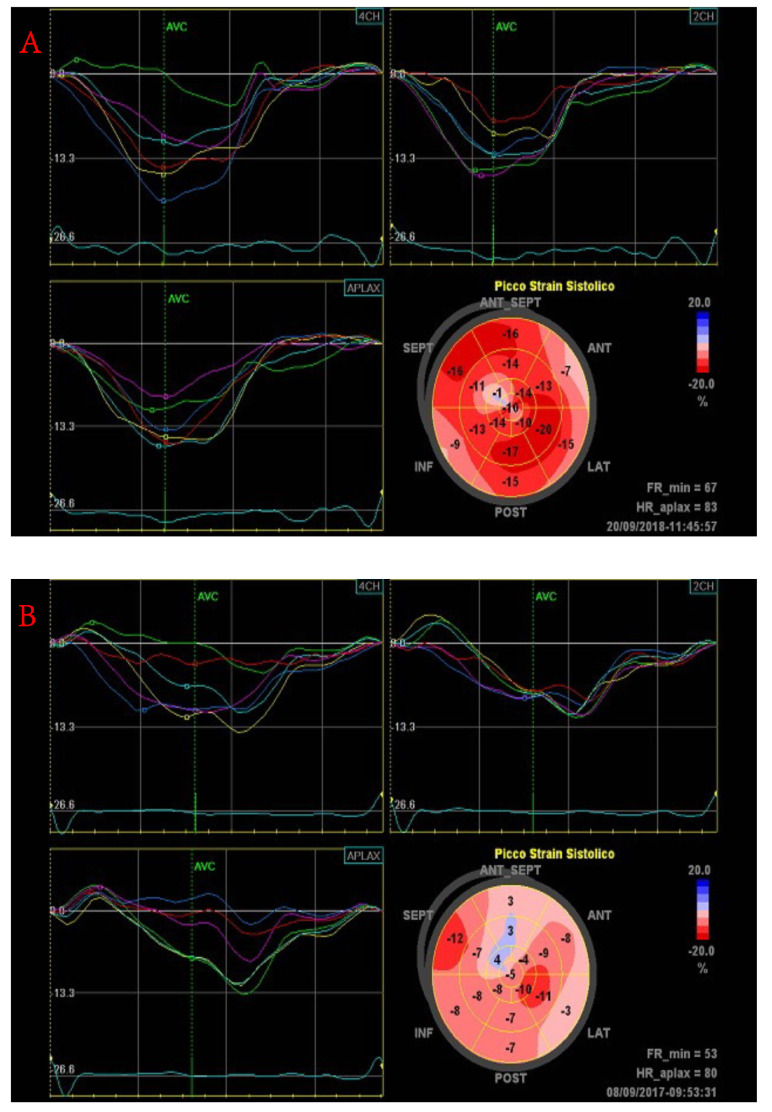
Evolution of global longitudinal strain in a patient with acute cellular rejection. Note the normal value at baseline (**A**) and the overall reduction during the rejection episode (**B**).

**Figure 5 diagnostics-11-02338-f005:**
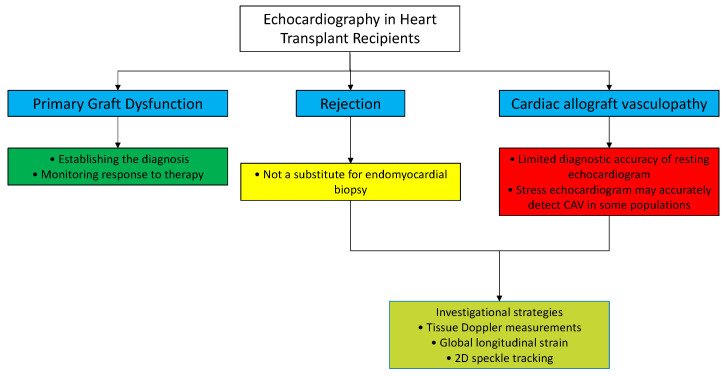
Role of echocardiography in HTRs. CAV: cardiac allograft vasculopathy.

**Table 1 diagnostics-11-02338-t001:** International Society Heart and Lung Transplantation (ISHLT) grading of acute cellular rejection.

ISHLT Grading	Grading of Rejection	Histopathological Findings
Grade 0	No rejection	No rejection
Grade 1 R	Mild	Interstitial and/or perivascular infiltrate with up to one focus of myocyte damage
Grade 2 R	Moderate	Two or more foci of infiltrates with associated myocyte damage
Grade 3 R	Severe	Diffuse infiltrate with multifocal myocyte damage, with or without edema, hemorrhage, or vasculitis

## Data Availability

Not applicable.

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
