# Peer review of "The Role of Echocardiography in the Management of Heart Transplant Recipients"

_diagnostics, 2021, doi:10.3390/diagnostics11122338_

Round 1

Reviewer 1 Report

Dear Authors,

first of all thank you for submitting your valuable paper.

The Authors analyzed the use of echocardiography in the heart transplant setting, an area in constant evolution thanks to technological progress in the field of echocardiographic techniques.

The paper appears well written and structured, the main fields of application are treated and there is a comparison with other diagnostic techniques that often still constitute the gold standard.

The usefulness and the current role of the method clearly emerges from the reading of the manuscript, just as its limitations appear sufficiently analyzed.

I just have some minor considerations for the Authors:

  • The use of 3D echocardiography and 3D STE methods is only hinted at in the conclusions: there are some works on these techniques (such as "Du GQ, Hsiung MC, Wu Y, Qu SH, Wei J, Yin WH, Tian JW. Three -Dimensional Speckle-Tracking Echocardiographic Monitoring of Acute Rejection in Heart Transplant Recipients. J Ultrasound Med. 2016 Jun; 35 (6): 1167-76 ") which could help complete the analysis.
  • A summary table of the histological degrees of graft rejection could be useful, since they are mentioned in the text.
  • There are some grammatical and typing errors to correct.

Author Response

We would like to thank the reviewer for his appreciation of our paper. 
In accordance with your comments we have
Carefully revised the paper to correct reported topographical and grammatical errors
Added a table on acute cellular rejection grading according to ISHLT criteria 
Regarding the role of 3 D echocardiography evaluation, we apologize to the reviewer but we believe that it is currently not usable in routine clinical practice and therefore we have decided not to cover it in our review which aims to provide practical messages on the use of echocardiography in HTRS.

Reviewer 2 Report

The manuscript deals with  clinically  important topic – echocardiography- primary non invasive modality in heart transplant recipients, as echocardiography  is a fundamental tool for adequately managing heart transplant recipients, from monitoring the immediate postoperative changes to identification of early and late post-transplant complications .

The manuscript presents comprehensive analysis of 72 publications and review of  echocardiographic features of  morphology and function of different chambers, valves and pericardium of the normal transplanted heart, as well as the  echocardiography tasks and findings in  primary graft dysfunction, acute graft rejection and allograft vasculopathy.        

 All the parts of the review are discussed thoroughly, including 5 figures. However,  it should be appreciable if the authors could present the protocol of echocardiographic investigation dedicated specifically for heart transplant recipients, noting what parameters should be evaluated obligatory.

.

The English level of the manuscript is high. The manuscript will be interesting to the wide auditorium of medical specialists, especially cardiologists, internal medicine specialists, etc.

I highly recommend the manuscript for publication, owing to very clinically relevant topic, consistent summary of key echocardiographic features of  the normal transplant heart and its pathological issues.

The manuscript deals with  clinically  important topic – echocardiography- primary non invasive modality in heart transplant recipients, as echocardiography  is a fundamental tool for adequately managing heart transplant recipients, from monitoring the immediate postoperative changes to identification of early and late post-transplant complications .

The manuscript presents comprehensive analysis of 72 publications and review of  echocardiographic features of  morphology and function of different chambers, valves and pericardium of the normal transplanted heart, as well as the  echocardiography tasks and findings in  primary graft dysfunction, acute graft rejection and allograft vasculopathy.        

 All the parts of the review are discussed thoroughly, including 5 figures. However,  it should be appreciable if the authors could present the protocol of echocardiographic investigation dedicated specifically for heart transplant recipients, noting what parameters should be evaluated obligatory.

.

The English level of the manuscript is high. The manuscript will be interesting to the wide auditorium of medical specialists, especially cardiologists, internal medicine specialists, etc.

I highly recommend the manuscript for publication, owing to very clinically relevant topic, consistent summary of key echocardiographic features of  the normal transplant heart and its pathological issues.

The manuscript deals with  clinically  important topic – echocardiography- primary non invasive modality in heart transplant recipients, as echocardiography  is a fundamental tool for adequately managing heart transplant recipients, from monitoring the immediate postoperative changes to identification of early and late post-transplant complications .

The manuscript presents comprehensive analysis of 72 publications and review of  echocardiographic features of  morphology and function of different chambers, valves and pericardium of the normal transplanted heart, as well as the  echocardiography tasks and findings in  primary graft dysfunction, acute graft rejection and allograft vasculopathy.        

 All the parts of the review are discussed thoroughly, including 5 figures. However,  it should be appreciable if the authors could present the protocol of echocardiographic investigation dedicated specifically for heart transplant recipients, noting what parameters should be evaluated obligatory.

.

The English level of the manuscript is high. The manuscript will be interesting to the wide auditorium of medical specialists, especially cardiologists, internal medicine specialists, etc.

I highly recommend the manuscript for publication, owing to very clinically relevant topic, consistent summary of key echocardiographic features of  the normal transplant heart and its pathological issues.

The manuscript deals with  clinically  important topic – echocardiography- primary non invasive modality in heart transplant recipients, as echocardiography  is a fundamental tool for adequately managing heart transplant recipients, from monitoring the immediate postoperative changes to identification of early and late post-transplant complications .

The manuscript presents comprehensive analysis of 72 publications and review of  echocardiographic features of  morphology and function of different chambers, valves and pericardium of the normal transplanted heart, as well as the  echocardiography tasks and findings in  primary graft dysfunction, acute graft rejection and allograft vasculopathy.        

 All the parts of the review are discussed thoroughly, including 5 figures. However,  it should be appreciable if the authors could present the protocol of echocardiographic investigation dedicated specifically for heart transplant recipients, noting what parameters should be evaluated obligatory.

.

The English level of the manuscript is high. The manuscript will be interesting to the wide auditorium of medical specialists, especially cardiologists, internal medicine specialists, etc.

I highly recommend the manuscript for publication, owing to very clinically relevant topic, consistent summary of key echocardiographic features of  the normal transplant heart and its pathological issues.

The manuscript deals with  clinically  important topic – echocardiography- primary non invasive modality in heart transplant recipients, as echocardiography  is a fundamental tool for adequately managing heart transplant recipients, from monitoring the immediate postoperative changes to identification of early and late post-transplant complications .

The manuscript presents comprehensive analysis of 72 publications and review of  echocardiographic features of  morphology and function of different chambers, valves and pericardium of the normal transplanted heart, as well as the  echocardiography tasks and findings in  primary graft dysfunction, acute graft rejection and allograft vasculopathy.        

 All the parts of the review are discussed thoroughly, including 5 figures. However,  it should be appreciable if the authors could present the protocol of echocardiographic investigation dedicated specifically for heart transplant recipients, noting what parameters should be evaluated obligatory.

.

The English level of the manuscript is high. The manuscript will be interesting to the wide auditorium of medical specialists, especially cardiologists, internal medicine specialists, etc.

I highly recommend the manuscript for publication, owing to very clinically relevant topic, consistent summary of key echocardiographic features of  the normal transplant heart and its pathological issues.

The manuscript deals with  clinically  important topic – echocardiography- primary non invasive modality in heart transplant recipients, as echocardiography  is a fundamental tool for adequately managing heart transplant recipients, from monitoring the immediate postoperative changes to identification of early and late post-transplant complications .

The manuscript presents comprehensive analysis of 72 publications and review of  echocardiographic features of  morphology and function of different chambers, valves and pericardium of the normal transplanted heart, as well as the  echocardiography tasks and findings in  primary graft dysfunction, acute graft rejection and allograft vasculopathy.        

 All the parts of the review are discussed thoroughly, including 5 figures. However,  it should be appreciable if the authors could present the protocol of echocardiographic investigation dedicated specifically for heart transplant recipients, noting what parameters should be evaluated obligatory.

.

The English level of the manuscript is high. The manuscript will be interesting to the wide auditorium of medical specialists, especially cardiologists, internal medicine specialists, etc.

I highly recommend the manuscript for publication, owing to very clinically relevant topic, consistent summary of key echocardiographic features of  the normal transplant heart and its pathological issues.

The manuscript deals with  clinically  important topic – echocardiography- primary non invasive modality in heart transplant recipients, as echocardiography  is a fundamental tool for adequately managing heart transplant recipients, from monitoring the immediate postoperative changes to identification of early and late post-transplant complications .

The manuscript presents comprehensive analysis of 72 publications and review of  echocardiographic features of  morphology and function of different chambers, valves and pericardium of the normal transplanted heart, as well as the  echocardiography tasks and findings in  primary graft dysfunction, acute graft rejection and allograft vasculopathy.        

 All the parts of the review are discussed thoroughly, including 5 figures. However,  it should be appreciable if the authors could present the protocol of echocardiographic investigation dedicated specifically for heart transplant recipients, noting what parameters should be evaluated obligatory.

.

The English level of the manuscript is high. The manuscript will be interesting to the wide auditorium of medical specialists, especially cardiologists, internal medicine specialists, etc.

I highly recommend the manuscript for publication, owing to very clinically relevant topic, consistent summary of key echocardiographic features of  the normal transplant heart and its pathological issues.

The manuscript deals with  clinically  important topic – echocardiography- primary non invasive modality in heart transplant recipients, as echocardiography  is a fundamental tool for adequately managing heart transplant recipients, from monitoring the immediate postoperative changes to identification of early and late post-transplant complications .

The manuscript presents comprehensive analysis of 72 publications and review of  echocardiographic features of  morphology and function of different chambers, valves and pericardium of the normal transplanted heart, as well as the  echocardiography tasks and findings in  primary graft dysfunction, acute graft rejection and allograft vasculopathy.        

 All the parts of the review are discussed thoroughly, including 5 figures. However,  it should be appreciable if the authors could present the protocol of echocardiographic investigation dedicated specifically for heart transplant recipients, noting what parameters should be evaluated obligatory.

.

The English level of the manuscript is high. The manuscript will be interesting to the wide auditorium of medical specialists, especially cardiologists, internal medicine specialists, etc.

I highly recommend the manuscript for publication, owing to very clinically relevant topic, consistent summary of key echocardiographic features of  the normal transplant heart and its pathological issues.

The manuscript deals with  clinically  important topic – echocardiography- primary non invasive modality in heart transplant recipients, as echocardiography  is a fundamental tool for adequately managing heart transplant recipients, from monitoring the immediate postoperative changes to identification of early and late post-transplant complications .

The manuscript presents comprehensive analysis of 72 publications and review of  echocardiographic features of  morphology and function of different chambers, valves and pericardium of the normal transplanted heart, as well as the  echocardiography tasks and findings in  primary graft dysfunction, acute graft rejection and allograft vasculopathy.        

 All the parts of the review are discussed thoroughly, including 5 figures. However,  it should be appreciable if the authors could present the protocol of echocardiographic investigation dedicated specifically for heart transplant recipients, noting what parameters should be evaluated obligatory.

.

The English level of the manuscript is high. The manuscript will be interesting to the wide auditorium of medical specialists, especially cardiologists, internal medicine specialists, etc.

I highly recommend the manuscript for publication, owing to very clinically relevant topic, consistent summary of key echocardiographic features of  the normal transplant heart and its pathological issues.

Author Response

We would like to thank the reviewer for his appreciation of our paper. 
In accordance with your comment, we have inserted a sentence clarifying that the graft evaluation must be based on the same elements as the native heart (therefore the echocardiographic minimum data set must also be the same), but that these elements must be interpreted in the pathophysiology of the transplanted heart

"The echocardiographic evaluation of the graft is based on the assessment of the same morphological and functional parameters that are considered for the native heart. However, the results of this evaluation must be interpreted taking into account the pathophysiology of the transplanted heart, therefore, the main morphological and functional aspects to be considered normal in HTRs will be briefly described."